# Attenuating Dynamics of Strongly Interacting Fermionic Superfluids in SYK Solvable Models

Tian-Gang Zhou[1] and Pengfei Zhang[2]*

**1** Institute for Advanced Study, Tsinghua University, Beijing,100084, China
**2** Department of Physics, Fudan University, Shanghai, 200438, China
* pengfeizhang.physics@gmail.com

March 22, 2023

## Abstract

Quench dynamics of fermionic superfluids is an active topic both experimentally and theoretically. Using the BCS theory, such non-equilibrium problems can be reduced to nearly independent spin dynamics only with a time-dependent mean-field pairing term. This results in persisting oscillations of the paring strength in certain parameter regimes. In experiments, however, it is observed that the oscillation decays rapidly when the interaction becomes strong, such as in the unitary fermi gas. A theoretical analysis is still absent. In this work, we construct an SYK-like model to analyze the effect of strong interactions in one dimensional BCS system. We utilize the large-$N$ approximation and Green's function-based technique to solve the equilibrium problem and quench dynamics. We find that a strong SYK interaction suppresses the paring order. We further verify that the system quickly thermalizes with SYK interactions for both intrinsic pairing order or proximity effect, which leads to a rapid decay of the strength of the oscillations. The decay rates exhibit different scaling laws against SYK interaction, which can be understood in terms of the Boltzmann equation. Our work makes a first step towards the understanding of attenuating dynamics of strongly interacting fermionic superfluids.

# 1 Introduction

Non-equilibrium dynamics in strong interaction systems is one of the most intriguing subjects in condensed matter systems and ultracold atoms. In particular, there is an increasing interest in understanding the quench dynamics, which means observing the evolution induced by a rapid parameter change. Decades ago, seminal works explore novel quench dynamics in superconductors [1–8]. Different dynamical phases are classified when adjusting the initial and final strength of attractive interactions according to the behavior of the paring strength. Persisting oscillation of the order parameter exhibits in the phase diagram. This is because the pioneering BCS theory can also be interpreted in the Anderson spin language. The oscillation can be interpreted as the collective mode of Anderson spins in the mean-field.

However, the Fermi superfluids realized in ultracold gases may not be in a collisionless regime if a magnetic field is used to tune the scattering length between atoms [9, 10]. In particular, the unitary fermi gas is the typical strongly interacting system that can be realized in the experiment [11–17]. Unfortunately, the theoretical treatments of quench dynamics in unitary fermi gas don't reach a consensus yet. For a simple trial, we consider adding extra interaction between Anderson spins in addition to the BCS type mean-field interaction. For simplicity, we treat these additional interactions as all-to-all and Gaussian random interactions, inspired by the famous exact solvable Sachdev-Ye-Kitaev model [18–37]. We assume the interaction is intra-spin and is independent for different spin componenets, which is different from models for the eternal traversable wormholes [38–50].

We analyze the effect of SYK-type interaction in terms of one dimension BCS system utilizing the large-$N$ approximation and Green's function-based technique for both the equilibrium problem and quench dynamics. We first study the superconductivity transition point by calculating the critical parameter hypersurface, and then the equilibrium phase diagram of pairing with finite order parameters or BCS interaction. These phase diagrams all demonstrate that SYK interaction weakens the superconductivity. We further numerically obtain the non-equilibrium quench dynamics, where the oscillation amplitude is suppressed by SYK interaction $J$, which is consistent with the equilibrium phase result. Finally, we notice that the decay rate exhibits different scaling laws against interaction $J$ depending on whether the paring is intrinsic or from the proximity effect. We argue that this can be understood in terms of the Boltzmann equation [51].

# 2 Model

Here we study the model in a one-dimensional spin-1/2 fermionic lattice model with up and down spin $l = \uparrow, \downarrow$. Depicted in Fig. 1, the Hamiltonian is composed of two parts: the one-dimensional BCS Hamiltonian and the intracell complex SYK-type interaction without any correlation between coupling constants for fermions with different spins. The Hamiltonian reads

$$\hat{H} = \hat{H}_{\text{BCS}} + \hat{H}_{\text{I}}, \tag{1}$$

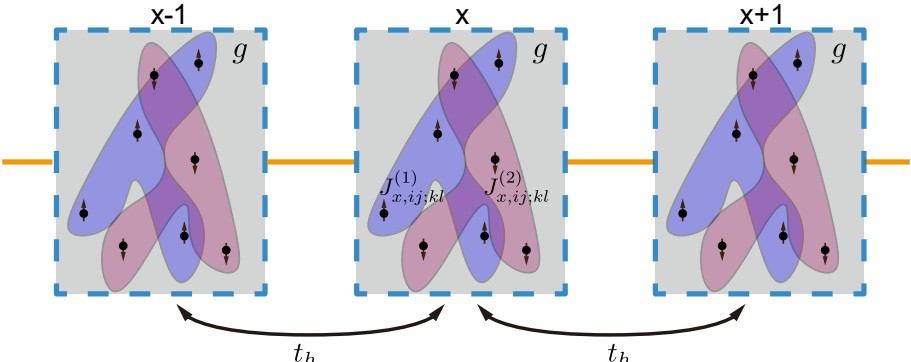

Figure 1: A pictorial representation of the model described by Eq. (1), (2) and (4). Different dots with arrows represent different spin 1/2 fermion in each unit cell. The blue and red blobs represent the intracell random interactions among the same spin and the solid lines with arrows denote the constant intercell hopping in the lattice. In this large-$N$ limit, the onsite attraction can be treated in the mean-field level, defined with interaction strength $g > 0$.

The BCS Hamiltonian is

$$\hat{H}_{\text{BCS}} = \sum_{p,i,l}(\epsilon_p - \mu)\hat{c}^\dagger_{p,i,l}\hat{c}_{p,i,l} - \sum_{p,i}(\Delta\hat{c}^\dagger_{p,i,\uparrow}\hat{c}^\dagger_{-p,i,\downarrow} + \Delta^*\hat{c}_{-p,i,\downarrow}\hat{c}_{p,i,\uparrow}). \quad (2)$$

Here $i = 1, 2, ..., N$ labels different modes on a single site for each spin flavor. As a 1-d model, we assume the band dispersion as $\epsilon_p = -2t_h\cos(p)$, where $t_h$ is the hopping strength, and $p \in (-\pi, \pi]$. $\mu = 0$ corresponds to half-filling because of the particle-hole symmetry after the disorder average. To proceed with equilibrium calculations, we need to distinguish two possible origins of order parameters $\Delta$. The type I comes from the background quantum proximity effect with a fixed order parameter, and the type II considers the self-consistent of the order parameter in evolution, which is

$$\Delta = g/N_s \sum_p \langle \hat{c}_{-p,i,\downarrow}\hat{c}_{p,i,\uparrow}\rangle_i. \quad (3)$$

Here we take the convention $g > 0$ to represent the attractive interaction strength. We define the intercell disorder average $\langle \cdots \rangle_i \equiv 1/N \sum_i^N \langle \cdots \rangle$, where $N_s$ is the number of sites. We take the large-$N$ limit in the later calculations and fix $N_s = 32$ when performing numerical simulations for the quench dynamics.

The intracell SYK-type interaction reads

$$\hat{H}_{\text{I}} = \sum_{x,i<j;k<l} J^{(1)}_{x,ij;kl}\hat{c}^\dagger_{x,i,\uparrow}\hat{c}^\dagger_{x,j,\uparrow}\hat{c}_{x,k,\uparrow}\hat{c}_{x,l,\uparrow}$$
$$+ \sum_{x,i<j;k<l} J^{(2)}_{x,ij;kl}\hat{c}^\dagger_{x,i,\downarrow}\hat{c}^\dagger_{x,j,\downarrow}\hat{c}_{x,k,\downarrow}\hat{c}_{x,l,\downarrow}, \quad (4)$$

where the random couplings in $\hat{H}_{\text{I}}$ obey the following relations

$$\begin{aligned}\text{expectation} \quad & \overline{J^{(1)}_{x,ij;kl}} = \overline{J^{(2)}_{x,ij;kl}} = 0 \\ \text{variance} \quad & \overline{J^{(1)}_{x,ij;kl}J^{(1)}_{x',ij;kl}} = \overline{J^{(1)}_{x,ij;kl}J^{(1)}_{x',ij;kl}} = 2J^2/N\delta_{xx'}.\end{aligned} \quad (5)$$

We have introduced two random couplings $J^{(1)}_{x,ij;kl}, J^{(2)}_{x,ij;kl}$, which correspond to uncorrelated random interactions for fermions with different spins. Recently, several works also proposed similar construction for the superconductivity Sachdev-Ye-Kitaev (SYK) model, but in different dimensionality or correlated SYK interactions. [52–54].

## 2.1 Methods for Thermal Equilibrium Properties

Following the standard approach elaborated in related works [24, 38, 55], we define the retarded Green's function in terms of Nambu spinor representation $(\hat{\psi}_{p,j,1}, \hat{\psi}_{p,j,2}) = (\hat{c}_{p,j,\uparrow}, \hat{c}^{\dagger}_{-p,j,\downarrow})$

$$
\begin{aligned}
G^{>}_{jj';ss'}(p;t,t') &\equiv -i\langle \hat{\psi}_{p,j,s}(t) \hat{\psi}^{\dagger}_{p,j',s'}(t')\rangle \delta_{jj'} \\
G^{<}_{jj';ss'}(p,t,t') &\equiv i\langle \hat{\psi}^{\dagger}_{p,j',s'}(t') \hat{\psi}_{p,j,s}(t)\rangle \delta_{jj'},
\end{aligned}
\tag{6}
$$

where $s = 1, 2$ represents two components of the Nambu spinor. In the thermal equilibrium, all Green's functions are only functions of $t - t'$ due to the time-translational symmetry, with $G^{\gtrless}_{jj';ss'}(p;t,t') = G^{\gtrless}_{ss'}(p, t-t')\delta_{jj'}$. The diagonal of intracell index $j$ in Green's function is due to the disorder average, and later on, we will ignore the intracell index $j$ in Green's function for convenience. Furthermore, we introduce the retarded Green's function $G^{R/A}$ related to $G^{\gtrless}$ as

$$
G^{R/A}_{ss'}(p;t,t') = \pm\Theta\left(\pm(t-t')\right)\left(G^{>}_{ss'}(p;t,t') - G^{<}_{ss'}(p;t,t')\right),
\tag{7}
$$

where $\Theta(t)$ is the Heaviside step function. By performing the Fourier transformation, Green's function can be represented on the momentum and frequency domain.

$$
G^{R/A}_{ss'}(p,\omega) = \int \mathrm{d}t\ G^{R/A}_{ss'}(p,t)e^{-i\omega t}.
$$

Then we can obtain the self-consistent Schwinger-Dyson equation for the retarded Green's function

$$
(G^{R})^{-1}(p,\omega) = ((G^{0})^{R})^{-1}(p,\omega) - \Sigma^{R}(\omega)
\tag{8}
$$

The bare Green's function corresponds to the BCS Hamiltonian

$$
((G^{0})^{R})^{-1}(p,\omega) = (\omega + i0^{+})\hat{\sigma}^{0} - \epsilon(p)\hat{\sigma}^{z} + \Delta_{i}\hat{\sigma}^{x}.
\tag{9}
$$

Here $\{\hat{\sigma}^{0}, \hat{\sigma}^{r}\}$, with $r = x, y, z$ are the Pauli matrix in the basis of Nambu spinor. In terms of Green's function, the equilibrium order parameter $\Delta_i$ could be separately written as

$$
\Delta_i = \begin{cases} \Delta_{i,0} & \text{(constant)} & \text{Type I,} \\ -\sum_p i g_i G^{<}_{12}(p;t,t)/N_s & \text{Type II.} \end{cases}
\tag{10}
$$

By taking the large-$N$ limit, and utilizing the tools of Keldysh contour [51], the self-energy on the time domain could be written as

$$
\begin{aligned}
\Sigma^{\gtrless}_{ss'}(t,t') &= \overset{x,t'}{\underset{s'}{\longrightarrow}}\ \overset{x,t}{\underset{s}{\longrightarrow}} \\
&= J^{2}G^{\gtrless}_{ss}(x=0;t,t')G^{\lessgtr}_{ss}(x=0;t',t)G^{\gtrless}_{ss}(x=0;t,t')\delta_{ss'} \\
&= \frac{1}{N_s^3} \sum_{p_1,p_2,p_3} J^{2}G^{\gtrless}_{ss}(p_1;t,t')G^{\lessgtr}_{ss}(p_2;t',t)G^{\gtrless}_{ss}(p_3;t,t')\delta_{ss'},
\end{aligned}
\tag{11}
$$

where $G^{\gtrless}_{ss}(x;t,t')$ is the Fourier transformation of $G^{\gtrless}_{ss}(p;t,t')$ and the retarded/advanced self-energy are similarly defined as

$$
\Sigma^{R/A}_{ss'}(t,t') = \pm\Theta\left(\pm(t-t')\right)\left(\Sigma^{>}_{ss'}(t,t') - \Sigma^{<}_{ss'}(t,t')\right).
\tag{12}
$$

We notice the self-energy Eq. (11) only has spin diagonal terms, since the coupling $J^{(1)}_{x,ij;kl}, J^{(2)}_{x,ij;kl}$ are not correlated. Besides, the $x = 0$ in the Green's function results from the intercell disorder average in the Eq. (5).

To solve the real-time Green's functions self-consistently, we introduce the spectral function as

$$G_{ss'}^R(p, \omega) = \int dz \frac{\rho_{ss'}(p, z)}{z - \omega + i0^+}, \tag{13}$$

which implies $\rho_{ss'}(p, \omega) = -\mathrm{Im} G_{ss'}^R(p, \omega)/\pi$. The greater and lesser Green's functions are associated with spectral function by the fluctuation-dissipation theorem as

$$
\begin{aligned}
G_{ss'}^<(p, \omega) &= 2\pi i n_F(\omega)\rho(p, \omega)_{ss'}, \\
G_{ss'}^>(p, \omega) &= -2\pi i n_F(-\omega)\rho(p, \omega)_{ss'},
\end{aligned}
\tag{14}
$$

where $n_F(\omega)$ is the Fermi-Dirac distribution function. By using Eq. (8) and Eq. (11), one can iteratively obtain the equilibrium spectral functions and Green's functions.

## 2.2 Methods for Non-equilibrium Dynamics

To study the quench dynamics, we choose the real-time approach and utilize the Kadanoff-Baym equation on the Keldysh contour [51], which describes the real-time evolution of $G^{\gtrless}$. Assuming the melon diagram approximation Eq. (11) and applying the Langreth rules [56] on the Schwinger-Dyson equation, we find that [38]:

$$
\begin{aligned}
i\partial_{t_1} G^{\gtrless}(p; t_1, t_2) + (-\epsilon(p)\hat{\sigma}^0 + \Delta_f(t_1)\hat{\sigma}^x) G^{\gtrless}(p; t_1, t_2) = \\
\int dt_3 (\Sigma^R(t_1, t_3) G^{\gtrless}(p; t_3, t_2) + \Sigma^{\gtrless}(t_1, t_3) G^A(p; t_3, t_2)), \\
-i\partial_{t_2} G^{\gtrless}(p; t_1, t_2) + G^{\gtrless}(p; t_1, t_2)(-\epsilon(p)\hat{\sigma}^0 + \Delta_f(t_2)\hat{\sigma}^x) = \\
\int dt_3 (G^R(p; t_1, t_3) \Sigma^{\gtrless}(t_3, t_2) + G^{\gtrless}(p; t_1, t_3) \Sigma^A(t_3, t_2)).
\end{aligned}
\tag{15}
$$

Similarly, we consider both the quantum proximity effect and self-consistent procedure of $\Delta(t)$. We summarize the two cases as

$$
\Delta_f(t > 0) = \begin{cases} \Delta_{f,0} & \text{(constant)} & \text{Type I,} \\ -\sum_p i g_f G_{12}^<(p; t, t)/N_s & \text{Type II.} \end{cases}
\tag{16}
$$

The quench protocol can be realized in the following manner. For $t_1, t_2 < 0$, we require that $G^{\gtrless}(p; t_1, t_2) = G^{\gtrless}(p, t_{12})$. In other words, $G^{\gtrless}(p; t_1, t_2)$ is given by the equilibrium solution with initial order parameter $\Delta_i$ defined in the Eq. (10) correspondingly, which serves as the initial conditions for the real-time dynamics. For $t_1, t_2 > 0$, the system drives away from the equilibrium with the new order parameter $\Delta_f(t)$. Solving the differential equation of $G^{\gtrless}(t_1, t_2)$ with the Eq. (11) and (15) gives the quantum dynamics. We apply the second order Euler's method and choose the time domain cutoff to be $t/\Delta t \in [-n_t, n_t]$ with $n_t = 2000$ and discrete time step $\Delta t = 20\beta/(n_t J)$. We have benchmarked the numerical error by testing the time translation invariance for Green's functions when we evolve the Green's functions without changing any parameters of the system.

## 3 Numerical Results

In this section, we present numerical results both in thermal equilibrium and for quench dynamics. In both cases, we find the paring strength is suppressed by the SYK random interactions. This qualitatively matches the observation in cold atom experiments.

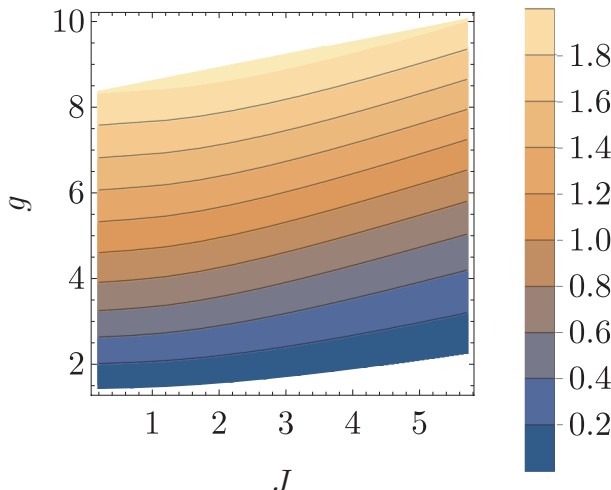

Figure 2: Critical BCS phase diagram with SYK type interaction. To illustrate the critical hyper-surface in the $g - T - J$ space, we project the hyper-surface into the $g - J$ plane using a contour plot. We fix the hopping strength $t_h = 0.1$.

## 3.1 Phase Diagram in Thermal Equilibrium

It is known that systems with attractive interactions exhibit superconducting transition at critical temperature $T_c$ for given BCS interaction strength $g$, or equivalently at critical BCS interaction strength $g_c$ at given temperature $T$. In this part, we aim to explore the effect of SYK interaction $J$ on the transition temperature $T_c$ by computing the equilibrium phase diagram for type II models. The phase diagram for the traditional BCS system can be restored by taking $J \to 0$.

The transition temperature $T_c$ can be determined by solving the gap equation with $\Delta = 0$ [57, 58]. Here we determine $T_c$ by taking a finite but small order parameter $\Delta = 10^{-3}$ and perform the iteration for Green's functions for a fixed $J$ in the limit of $N_s \to \infty$ (see appendix A for the details in taking the limit). After the Green's functions converge, $g$ is computed by using the relation (10) (type II). In numerics, we fix the hopping strength $t_h = 0.1$. In fig. 2, we show critical hypersurface in the $g - T - J$ space through a contour plot on the $g - J$ plane. We find with fixed $T$, larger $J$ leads to larger critical BCS interaction $g_c$. Since the superconductivity happens when the BCS interaction is larger than $g_c$, it indicates that the SYK interaction weakens the superconductivity.

We further compute the pairing strength for different SYK interaction strength $J$ with finite order parameter $\Delta$ or BCS interaction $g$ in type I and type II systems correspondingly. We define the pairing strength $\alpha \in (-0.5, 0.5)$ as a 'normalized' order parameter, which reads

$$\alpha \equiv 1/N_s \sum_p \langle \hat{c}_{-p,i,\downarrow} \hat{c}_{p,i,\uparrow} \rangle_i. \tag{17}$$

It corresponds to observing the magnetization in $x$ direct in the language of Anderson's pseudospin [2]. This also indicates the close relation between the attenuating dynamics of fermionic superfluids and the magnetization dynamics of the random spin model [38, 59]

The fig.3 (a), (b) show equilibrium pairing $\alpha$ in type I and type II system respectively. There are two remarks on the results. First, the proximity effect leads to a smooth change in the non-zero pairing $\alpha$ against the fixed order parameter $\Delta$. This can be understood as the magnetization induced by an external traverse magnetic field in the Anderson spin model, which is always non-zero. As a comparison, with BCS self-consistency, there is a typical second-order phase transition phenomenon at $g = g_c$ with each SYK interaction $J$. This is consistent

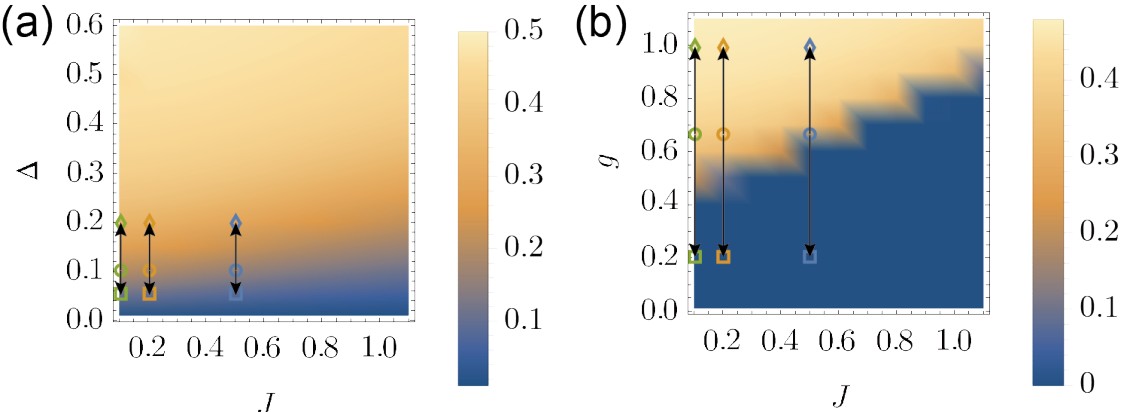

Figure 3: The pairing $\alpha$ in the type I and II system. The color of the heatmap indicates the value of $\alpha$. We fix $t_h = 0.1$, $\beta = 10$ and choose finite $N_s = 32$. (a) Type I system with proximity effect. The open markers label the initial order parameter $\Delta_i = 0.1$ (circle) and quench to the final order parameter $\Delta_f = 0.1, 0.2$ (square and diamond). (b) Type II system with BCS iteration. The open markers label the initial BCS interaction $g_i = 0.66$ (circle) and quench to the two sets of final BCS interactions $g_f = 0.2, 1.0$ (square and diamond). Green, orange, and blue colors indicate SYK interaction $J = 0.1, 0.2, 0.5$ separately.

with the original BCS theory [58,60]. Secondly, here we choose finite discretization of momentum $N_s = 32$, for benchmarking the later calculation of quench dynamics. However, we find the $N_s = 32$ result in fig. 3(b) still qualitatively agree with the phase diagram illustrated in fig. 2 obtained in the limit of $N_s \to \infty$. Both of them show a positive correlation between critical $g$ and SYK interaction $J$. It provides a check for the validity of the finite $N_s$ calculations for the quench dynamics in the following subsection.

## 3.2 Attenuating Dynamics

In the limit of $J \to 0$, our model is equivalent to the standard BCS mean-field theory. Using the time-dependent Bogoliubov theory, previous literature shows that small-amplitude oscillations of order parameter persist with a frequency of $2\Delta$, which is the energy of the Higgs mode [1, 61–63]. Later studies propose phase diagram with three dynamical phases classified according to the dynamics of the order parameter after a quantum quench. [3–6]. The order parameter can disappear rapidly, damply oscillate, or persistently oscillate. However, these conclusions are obtained at the limit of large $N_s$ and assume a constant density of state. Here we can only take small $N_s$ and free fermion lattice dispersion in Eq. (2), limited by the Green's function-based numerical method. We hereby focus on the dependence of the decaying rate on the SYK interaction parameter $J$.

In fig. 3, we mark several open markers as the initial and final parameters for the quench dynamics numerics. We choose two sets of parameters: one set quenches to the system with a large order parameter (circle to diamond), and another one quenches to the system with a smaller order parameter(circle to square). In practice, it is realized by quenching superconducting order $\Delta$ induced by proximity effect in type I system, and by quenching BCS interaction $g$ in type II system. Besides, we are interested in the effect of SYK interaction $J$ on the non-equilibrium dynamics. Therefore, we also mark $J = 0.1, 0.2, 0.5$ in the equilibrium phase diagram for comparison in the quench dynamics.

The fig. 4 shows the oscillation is pervasive in different parameter regions. However, our

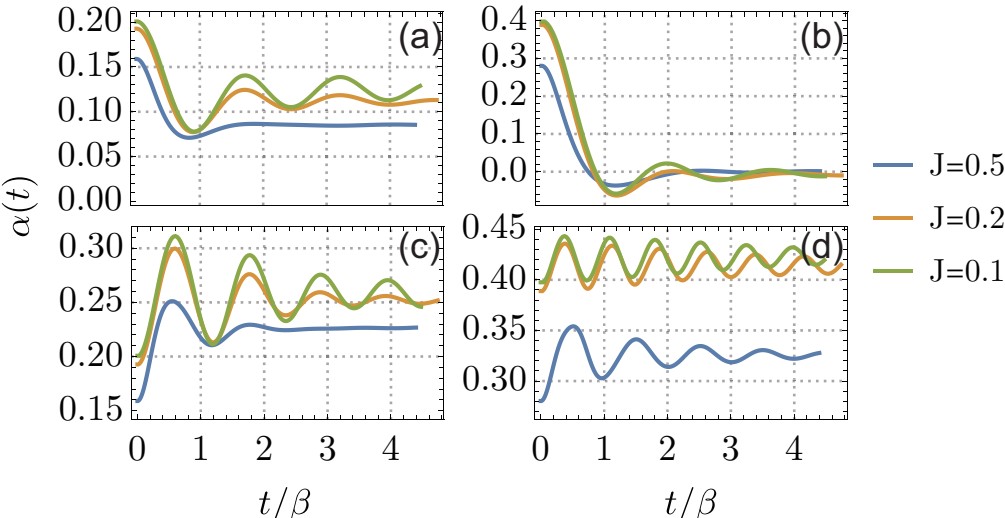

Figure 4: Quench dynamics of pairing in two types of system. We also fix $t_h = 0.1$, $\beta = 10$ and choose finite $N_s = 32$ according to fig. 3. (a, c) Quench dynamics with proximity effect. The system is initially prepared at $\Delta_i$ and quenched to $\Delta_f$. (a): $(\Delta_i, \Delta_f) = (0.1, 0.05)$; (c): $(\Delta_i, \Delta_f) = (0.1, 0.2)$. (b, d) Quench dynamics with order parameter self-consistency. The interaction strength is initially $g_i$ and then quench to $g_f$. (b): $(g_i, g_f) = (0.66, 0.2)$; (d): $(g_i, g_f) = (0.66, 1.0)$.

numerical results show that the SYK-type interaction strongly attenuates the oscillation amplitude compared to the traditional BCS system. This qualitatively matches the absence of oscillation observed in the BCS-BEC quench experiment [9]. Here we discuss the result of quench dynamics both with or without order parameter self-consistency. We fix hopping $t_h = 0.1$ and inverse temperature $\beta = 10$ but leave $J$ as an adjustable parameter. From fig. 4, we simulate the quench dynamics with different $J$ and different iterative types. Fig. 4(a), (c) belongs to the type I system which quenches the background proximity order parameter, whereas Fig. 4(b), (d) represents the type II system which quenches the BCS interaction strength. For both types of systems, we find that when SYK-type interaction $J$ increases, the amplitude of the oscillation decreases. We recall that such decreasing in amplitude is consistent with the equilibrium phase diagram in fig. 3, which indicates that the SYK interaction weakens the superconductivity.

It's worth exploring the decay rate against the SYK interaction. We fit the decay rate $\Gamma$ with formula $\alpha(t) \sim \alpha_0 e^{-\Gamma t} \cos(\Omega t + \theta) + c$, and the detail parameters are left to the appendix B. As shown in fig. 5, we find type I and II system exhibits different scaling law for $J \gtrsim t_h$. The system with proximity effect shows perfect linear law, while the self-consistent BCS system shows quadratic scaling law. We argue this can be understood by a semi-classical Boltzmann equation in the limit of $t_h \to 0$ [58]. We start with the type I model describing the proximity effect. Given an order parameter $\Delta_f$, the system consists of Bogoliubov particles with energy $E_k = |\Delta_f|$. Without the SYK interaction $J$, the lifetime of the Bogoliubov particles is infinite. The quantum state after the quench can be viewed as a non-equilibrium state of Bogoliubov particles. The relaxation of $\alpha$ is then because of the decay of Bogoliubov particles induced by the SYK interactions. Under the semi-classical approximation, this can be estimated by

$$
\begin{aligned}
\Gamma_k = {} & 2\pi J^2 \int \frac{dk_2 dk_3 dk_4}{(2\pi)^3} \delta(E_k + E_{k_2} - E_{k_3} - E_{k_4}) \\
& \times \left( n_F(E_{k_3}) n_F(E_{k_4})(1 - n_F(E_{k_2})) + (1 - n_F(E_{k_3}))(1 - n_F(E_{k_4})) n_F(E_{k_2}) \right).
\end{aligned}
\tag{18}
$$

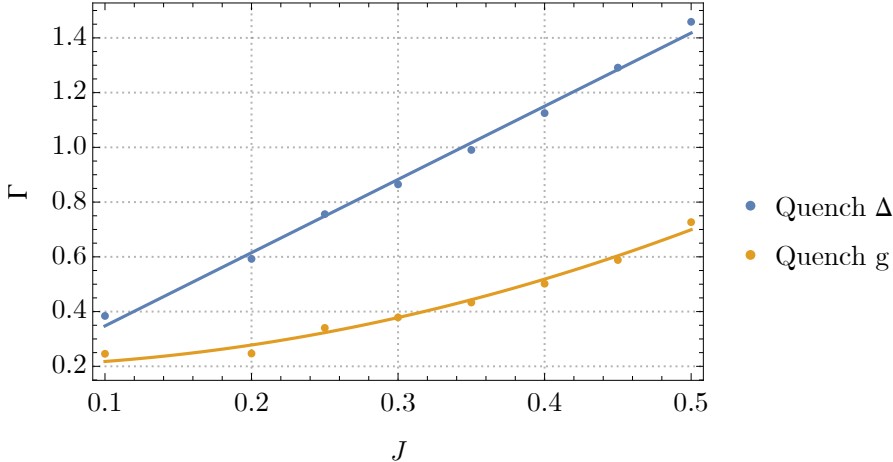

Figure 5: The decay rate in terms of different SYK interaction $J$. We also fix $t_h = 0.1$, $\beta = 10$ and choose finite $N_s = 32$. The initial and final conditions correspond to fig. 4(c) in the *Quench* $\Delta$ line, and correspond to fig. 4(d) in the *Quench g* line.

Here we focus on a two-to-two scattering for concreteness. Other scattering channels leads to a similar contribution. Unfortunately, the delta function diverges since $E_k = |\Delta_f|$. We expect this should be regularized as $\delta(0) \rightarrow 1/\Gamma$. As a result, we find $\Gamma \sim J^2/\Gamma$, which indicates $\Gamma \propto J$. A similar phenomenon appears in the high-temperature limit of the Majorana SYK model [64]. On the other hand, for the type II model, the order parameter $\Delta$ is dynamical. As a result, the instantaneous spectral function of fermions in non-equilibrium dynamics is generally continuous in time. This indicates the lifetime of quasi-particles can be finite even without $J$. If this is the case, we expect the contribution from finite $J$ takes the form of $\Gamma \sim \Gamma_0 + J^2/\Gamma_0$, which explains the quadratic dependence of $\Gamma$ with respect to $J$.

## 4 Discussion

In this work, we analyze the effect of SYK interactions in one dimension BCS system. We utilize the large-$N$ approximation and Green's function-based technique to solve the equilibrium problem and quench dynamics. We first calculate the critical hypersurface in $g - T - J$ parameter space which denotes the superconductivity transition. We also study the equilibrium phase diagram of pairing with finite order parameters or BCS interaction. These phase diagrams all demonstrate that SYK interaction suppresses the superconductivity order. With the help of the phase diagram, we explore the non-equilibrium quench dynamics. We find the oscillation of the paring strength is damped by SYK interaction $J$, consistent with the equilibrium phase diagram. This is understood as the introduction of interaction between Anderson spins, which physically thermalizes the Anderson spin system.

Our results serve as the first step towards a complete understanding of the attenuating dynamics of strongly interacting superconductors (or fermionic superfluids). For example, it is plausible to expect the unitary fermi gas also thermalizes quickly due to strong interactions between atoms [9]. Consequently, the paring strength should relax much more rapidly compared to results obtained in the traditional BCS theory. We defer the development of a microscopic description for the quench dynamics in the unitary fermi gas at low temperatures to future study.

## Acknowledgements

We are especially grateful for the invaluable discussions with Hui Zhai, whose advice is indispensable for the whole work.

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

# A   Critical parameter obtained in $N_p \to \infty$ limit

In this appendix, we show the technique to realize $N_p \to \infty$ in equilibrium calculation. We take care of the whole band dispersion and perform integration on the momentum space.

Recalling the Schwinger-Dyson equation (8), (9), now we will perform integration on the momentum index.

$$
G^R(x=0, \omega) = \int_{-\pi}^{\pi} \frac{\mathrm{d}p}{2\pi} \begin{pmatrix} -\epsilon(p) - \Sigma_{11}^R(\omega) + \omega + i0^+ & \Delta \\ \Delta & \epsilon(p) - \Sigma_{11}^R(\omega) + \omega + i0^+ \end{pmatrix}^{-1} \quad (19)
$$

Here we use the symmetry of Green's function, which is referred to previous work [38,59]

$$
G_{s_1 s_2}^>(t_1, t_2) = \begin{pmatrix} G_{22}^>(t_1, t_2) & G_{21}^>(t_1, t_2) \\ G_{12}^>(t_1, t_2) & G_{11}^>(t_1, t_2) \end{pmatrix}_{s_1 s_2} \quad (20)
$$

$$
G_{s_1 s_2}^>(t_1, t_2) = \begin{pmatrix} -G_{11}^<(t_2, t_1) & G_{12}^<(t_2, t_1) \\ G_{12}^<(t_2, t_1) & -G_{11}^<(t_2, t_1) \end{pmatrix}_{s_1 s_2}. \quad (21)
$$

Therefore we only need to consider the 11 and 12 components of the Green's function and self-energy. Also we remember $\Sigma_{12}^R = 0$ due to the format of SYK interaction Eq. (5).

Integrate over momentum leads to the final result

$$
\begin{aligned}
G^R(x=0, \omega)_{11} &= -\int_{-\pi}^{\pi} \frac{\mathrm{d}p}{2\pi} \frac{\epsilon(p) + \omega - \Sigma_{11}^R(\omega)}{\epsilon(p)^2 + \Delta^2 - \left(\omega - \Sigma_{11}^R(\omega)\right)^2} \\
&= -\frac{\omega - \Sigma_{11}^R(\omega)}{\sqrt{A(A + 4t^2)}} (-1)^{\mathrm{Floor}\left[\frac{\pi + \mathrm{Arg}[A + 4t^2] - \mathrm{Arg}[A]}{2\pi}\right]},
\end{aligned} \quad (22)
$$

and

$$
\begin{aligned}
G^R(x=0, \omega)_{12} &= \int_{-\pi}^{\pi} \frac{\mathrm{d}p}{2\pi} \frac{\Delta}{\epsilon(p)^2 + \Delta^2 - \left(\omega - \Sigma_{11}^R(\omega)\right)^2} \\
&= \frac{\Delta}{\sqrt{A(A + 4t^2)}} (-1)^{\mathrm{Floor}\left[\frac{\pi + \mathrm{Arg}[A + 4t^2] - \mathrm{Arg}[A]}{2\pi}\right]},
\end{aligned} \quad (23)
$$

where the polynomial $A = \Delta^2 - \left(\omega - \Sigma_{11}^R(\omega)\right)^2$. We have already known that self-energy only depends on Green's function located at $x = 0$. Hence, the integrated Schwinger-Dyson equation gives rise to a close form and can be solved self-consistently.

# B   Details of fitting decay rate

Different scaling laws of decay rate in quench $\Delta$ and quench $g$ protocols are revealed in fig. 5. Here we fit each $\alpha(t)$ curve with the first 280 data points for both quench protocols, as shown in fig. 6. The fitting formula is

$$
\Gamma = \begin{cases} 0.0799397 + 2.67588J & (\text{Quench } \Delta) \\ 0.197641 + 2.00575J^2 & (\text{Quench } g) \end{cases}, \quad (24)
$$

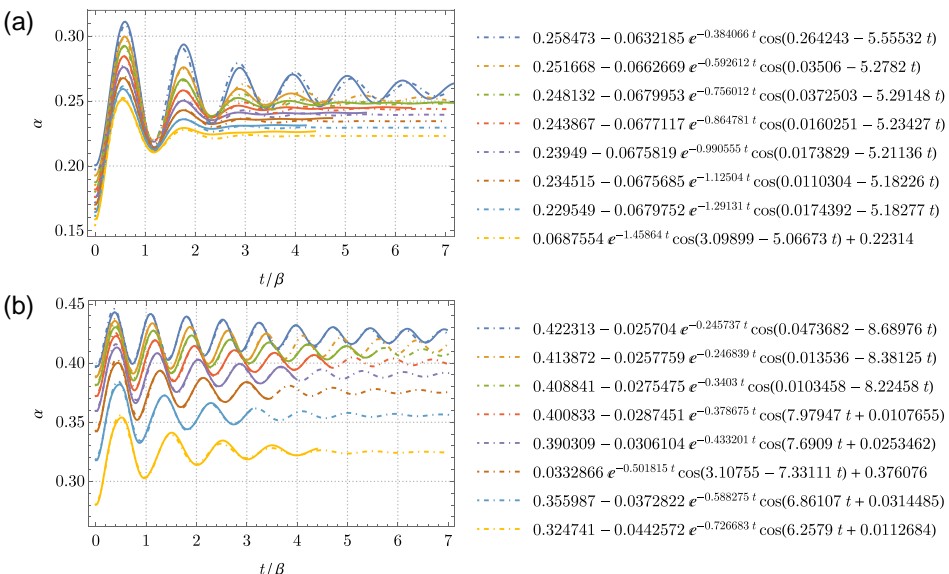

Figure 6: The fitting of decay rate in terms of different SYK interaction $J$. We fix $t_h = 0.1$, $\beta = 10$ and choose finite $N_s = 32$. The initial and final conditions of (a) correspond to *Quench* $\Delta$ line in the fig. 5, and (b) corresponds to the *Quench g* line of fig. 5. The right panel illustrates the detailed fitting formulas. From top to bottom, each curve corresponds to different SYK interactions $J = 0.1, 0.2, 0.25, 0.3, 0.35, 0.4, 0.45, 0.5$.

which is obtained by Mathematica `FindFit` formula.

For concreteness, we have tested the robustness of our conclusion by adjusting different fitting time periods in the data. There is no qualitative difference between different fitting regions.