# Peer review of "Attenuating Dynamics of Strongly Interacting Fermionic Superfluids in SYK Solvable Models"

_SciPost Physics_

## Round 1 · Referee Report · Anonymous (Referee 1) · 2023-6-22

Strengths
1. Interesting and timely
2. Quite well written and defining fairly precise goals
3. The overall logic is easy to follow and the description of the results is quite clear and useful
Weaknesses
1. Stronger motivation to consider their specific models
2. For non-expert readers, referencing could be more complete
Report
This manuscript discusses a timely subject and deals with computable models that allow the authors to derive new interesting results that bring the community to a better understanding in the set of open problems discussed in the introduction. For these reasons, my recommendation would be to accept it for publication in this journal after the requests below are taken care of.
Requested changes
1. Despite the motivation given in the introduction, some readers may end up with the impression the main reason the authors considered this model is because it provides a potentially solvable model (in some regime of parameters and energies) where one is probing the effect of some interaction on the degrees of freedom governed by the BCS action. It would benefit the readership of the manuscript if the authors have any stronger reasons to consider these models, in particular, the specific choice appearing in equation (4).
2. It would appear that when they first define the average N should be replaced with $N_s$
3. When applying the large N limit to the non-equilibrium dynamics, the authors mentioned the melodic approximation, i.e. the fact that equation (11) is the leading contribution is an assumption. Could they make this point more precise to help the readers appreciate the dynamical difference between the original equation (11) and the one being discussed in this context ?
4. Readers would benefit if the authors would add some references for : (a) the claim their numerical results qualitatively match similar observations in cold atom experiments at the start of section 3 and (b) the first statement in the first paragraph of section 3.1
5. When discussing the attenuating dynamics, can the authors provide any argument as to why they expect $\delta(0)$ to behave like $1/\Gamma$ ?
Anonymous on 2023-05-29 [id 3692]
This manuscript studies the coupling of a BCS system with an SYK-like model describing random interactions of fermions with different spins in order to gain some understanding and intuition for the attenuating dynamics observed in strongly interacting superconductors or fermionic superfluids.
The manuscript is quite well written and defining fairly precise goals. Once these are established, the overall logic is easy to follow and the description of the results is quite clear and useful.
I have some remarks and questions : - Despite the motivation given in the introduction, some readers may end up with the impression the main reason the authors considered this model is because it provides a potentially solvable model (in some regime of parameters and energies) where one is probing the effect of some interaction on the degrees of freedom governed by the BCS action. It would benefit the readership of the manuscript if the authors have any stronger reasons to consider these models, in particular, the specific choice appearing in equation (4). - It would appear that when they first define the average N should be replaced with $N_s$ - When applying the large N limit to the non-equilibrium dynamics, the authors mentioned the melonic approximation, i.e. the fact that equation (11) is the leading contribution is an assumption. Could they make this point more precise to help the readers appreciate the dynamical difference between the original equation (11) and the one being discussed in this context ? - Readers would benefit if the authors would add some references for : (a) the claim their numerical results qualitatively match similar observations in cold atom experiments at the start of section 3 and (b) the first statement in the first paragraph of section 3.1 - When discussing the attenuating dynamics, can the authors provide any argument as to why they expect $\delta(0)$ to behave like $1/\Gamma$
I believe the subject of this manuscript is interesting and timely, containing new results related to an interesting open problem. For these reasons, my recommendation would be to accept it for publication in this journal after the above requests are taken care of.

---

## Round 1 · Referee Report · Anonymous (Referee 2) · 2023-6-26

Strengths
1-Noval and Robust Approach: the paper innovatively uses an SYK-like model to address the question of rapidly decaying oscillations in strongly interacting fermionic superfluids, a question that has been challenging to tackle theoretically. The use of large-N approximation combined with Green's function-based technique ensures a robust and comprehensive approach to solve the equilibrium problem and quench dynamics.
2-Relevant to Experiments: The findings from this study are consistent with experimental observations of the rapid decay of oscillations when the interaction in a superfluid becomes strong.
Weaknesses
1-Interpretation of Calculations: Some of the calculation results presented in the study lack a thorough physical interpretation. This lack of context can make it challenging for readers to fully grasp the significance of these results or to apply them effectively to their own work. Providing more comprehensive explanations of the implications of the calculations would add depth to the analysis and could offer additional insights into the dynamics of the system under study.
Report
This paper achieves several of these expectations:
1-Breakthrough on Long-Standing Research Stumbling Block: The rapid decay of pairing strength oscillations in strongly interacting fermionic superfluids has been a long-standing issue in the field. This paper presents a significant breakthrough by providing a theoretical explanation for this phenomenon using an SYK-like model and a Green's function-based technique.
2-Opening New Pathways in Existing Research: By revealing the effects of strong SYK interactions on pairing order and the rate of thermalization, the paper opens new pathways for future research into the dynamics of strongly interacting fermionic superfluids.
3-Synergetic Link Between Research Areas: The use of an SYK-like model, originally developed in the context of quantum chaos and black hole physics, to analyze superfluid dynamics represents a novel and synergetic link between different research areas. This innovative application could potentially inspire further cross-disciplinary research.
Therefore, I recommend this paper to publish.
It will be better if the authors consider to give more interpretations of the numerical results. For instance, is the attenuated pairing related to the decreasing quasi-particle weight with the presence of the SYK coupling?
Requested changes
Please check spelling and grammar, e.g. several "pairing" is spelled as "paring"

---

## Editorial Decision

unknown